# Effect of Commonly Used Cosmetic Preservatives on Healthy Human Skin Cells

**DOI:** 10.3390/cells12071076

**Published:** 2023-04-03

**Authors:** Patrycja Głaz, Agata Rosińska, Sylwia Woźniak, Anna Boguszewska-Czubara, Anna Biernasiuk, Dariusz Matosiuk

**Affiliations:** 1Department of Synthesis and Chemical Technology of Pharmaceutical Substances, Medical University of Lublin, 20-093 Lublin, Poland; 2Department of Medical Chemistry, Medical University of Lublin, 20-093 Lublin, Poland; 3Department of Pharmaceutical Microbiology, Medical University of Lublin, 20-093 Lublin, Poland

**Keywords:** cosmetic preservatives, matrix metalloproteinase-2 (MMP-2) activity, human dermis, collagen secretion, cytotoxicity, human fibroblasts (BJ), human keratinocyte (HaCaT), antimicrobial activity

## Abstract

Cosmetic products contain preservatives to prevent microbial growth. The various types of preservatives present in skincare products applied on the skin induce many side effects. We tested several types of preservatives such as phenoxyethanol, methyl paraben, propyl paraben, imidazolidinyl urea (IU), the composition of gluconolactone and sodium benzoate (GSB), diazolidinyl urea (DU), and two grapefruit essential oils, one of which was industrially produced and a second which was freshly distilled from fresh grapefruit peels. This study aimed to find the relationship between preservative concentration, cell growth, collagen secretion, and cell viability. We hypothesized that these products induced a decrease in collagen secretion from human dermal fibroblasts. Our research, for the first time, addressed the overall effect of other preservatives on skin extracellular matrix (ECM) by studying their effect on metalloproteinase-2 (MMP-2) activity. Except for cytotoxicity and contact sensitivity tests, there are no studies of their effect on skin ECM in the available literature. These studies show potential antimicrobial activity, especially from the compounds IU and DU towards reference bacteria and the compounds methyl paraben and propyl paraben against reference fungi. The MTS test showed that fibroblasts are more sensitive to the tested group of preservatives than keratinocytes, which could be caused by the differences between the cells’ structures. The grapefruit oils exhibited the most cytotoxicity to both tested cell lines compared to all considered preservatives. The most destructive influence of preservatives on collagen synthesis was observed in the case of IU and DU. In this case, the homemade grapefruit oil turned out to be the mildest one. The results from a diverse group of preservatives show that whether they are natural or synthesized compounds, they require controlled use. Appropriate dosages and evaluation of preservative efficacy should not be the only aspects considered. The complex effect of preservatives on skin processes and cytotoxicity is an important topic for modern people.

## 1. Introduction

The permeation of substances through the skin is a complex process. It is influenced by the condition of the skin, which consists of hydration, the condition of the stratum corneum, or the tightness of the hydrolipid barrier, but also by the physicochemical properties of the substances in contact with it [1]. Preservatives are a large group of compounds used as additives in cosmetics, pharmaceuticals, and food. They show antimicrobial activity and prevent microbial growth in final products. However, the various types of preservatives present in skincare products induce many side effects such as skin irritation or contact dermatitis [2,3,4]. The alkyl esters of p-hydroxybenzoic acid (parabens) are widely used for preservation in cosmetics, food, and pharmaceuticals against microbial and fungal attacks [5]. Parabens are suitable for cosmetic formulations due to their physical properties: odorless, neutral pH, and not having any influence on cosmetic mass color. Parabens are also one of the best tested preservatives [6]. Phenoxyethanol, or 2-phenoxyethanol, has a wide spectrum of antimicrobial activity and is effective against various Gram-negative and Gram-positive bacteria and fungi belonging to yeasts. Phenoxyethanol is commonly used as a preservative in cosmetic products [7]. Diazolidinyl urea (germall II, DU) and imidazolidinyl urea (germall 115, IU) belong to the class of formaldehyde-releasing agents. They release formaldehyde during the entire durability of the product [2]. Furthermore, essential oils are an interesting field of study in pharmacy, medicine, and cosmetology due to their antimicrobial properties against a range of organisms. They are a composition of terpenes and sometimes phenylpropane. The essential oils of citrus peels contain D-limonene, α-terpinolene, β-myrcene, α- and β-pinene, γ-terpinene, α-caryophyllene, copaene, β-phellandrene, etc. [8,9,10,11]. Citrus peels contain flavonoids, and their antimicrobial properties are well-defined [12].

Skin as an external organ is exposed to many environmental factors, including cosmetic products, which can be the reason for non-immunological, chemically induced skin irritation. The in vivo skin reaction to chemical factors includes complex processes by direct contact between the skin and the irritant as well as the response of other organs. This complexity is difficult to match in an in vitro model as the skin is composed of many cell types such as keratinocytes, fibroblasts, melanocytes, Langerhans cells, mast cells, and Merkel cells. In vitro testing can use various skin cell lines as a good method of scanning many substances. Keratinocytes and fibroblasts are mostly used as representative cells of the epidermis and dermis. Keratinocytes which are developed at the air–liquid interface form a multilayered epidermis. The structure, which imitates a native epidermis, shows cellular layers, including a stratum basale, spinosum granulosum, and stratum corneum. At the ultrastructural level, hemidesmosomes, desmosomes, and keratohyalin granules can be observed [3]. Fibroblasts are the major mesenchymal cell type in connective tissue and deposit the collagen and elastic fibers of the extracellular matrix (ECM) [13]. Human skin dermis contains fibroblasts, which produce and adhere to the dermal ECM composed primarily of type I collagen fibrils [14]. That adherence and production of matrix metalloproteinases (MMPs) allow the fibroblasts to spread and exert mechanical force on the surrounding ECM and therefore maintain the proper composition and structural organization of the dermal ECM [15]. Collagen is the most widespread structural protein in the human body, forming molecular structures that strengthen tendons and elastic fibers that support the stability of skin structures, various tissues, and internal organs [16]. The mechanical properties of collagen are high stiffness (elastic modulus of 1 GPa) and toughness (tensile strength of 50–100 MPa) [17]. The alpha 1 (III) collagen chain (alpha 1 chain of type III collagen) is a protein that, in the human body, is encoded by the *COL3A1* gene. Three alpha 1 chains are required to form a type III collagen molecule, with a long triple helix domain. Type III collagen is an extracellular matrix protein, synthesized as pre-procollagen. It is a primary structural component in human organs such as large blood vessels, the uterus, and the intestines [18]. Among all collagens, type I and III collagens are the most commonly spotted, accounting for 70% and 5–20% of all collagens in mammals, respectively. In addition, type I and III collagens account for 80–85% and 10–15% of the total collagen in human skin, respectively [17].

MMPs are a family of around 30 (23 in humans) proteolytic enzymes whose role is to destroy ECM during normal tissue remodeling as a part of many biological processes in a healthy and pathological state. It was previously believed that these enzymes acted only to degrade components of the ECM, but more recent works on the identification of specific matrix and non-matrix substrates for MMPs elucidated their role more accurately in modulating normal cellular behavior and cell–cell communication as well as tumor progression processes [19]. Due to differences in their structure of the quaternary protein chain and substrate specificity, MMPs can be divided into six groups: collagenases (MMP-1, -8, and -13), gelatinases (MMP-2 and -9), stromelysins (MMP-3, -10, and -11), matrilysins (MMP-7 and -26), membrane MMPs (MMP-14, -15, -16, -17, -24, and -25), and unclassified MMPs. Although their substrates differ, all of them influence collagen degradation—i.e., collagenases could degrade collagen in fibrous form, while gelatinases are characterized by the highest activity against denatured collagen forms. The substrate characteristics of MMPs are quite wide-ranging—i.e., MMP-2 substrates are type I, II, III, IV, VII, IX, X, and XI collagen; gelatin; aggrecan; elastin; fibronectin; IGFBP; fibrine; fibrinogen; pro-MMP-9; pro-MMP-13; plasminogen; and pro-TNFα [13,20,21]. They perform a wide range of roles in physiological processes, which is the reason for their strict regulation by numerous mechanisms including natural tissue inhibitors of metalloproteinases (TIMPs). Research has only started to discover the more troublesome aspects of MMPs’ function, such as cancer progression, Alzheimer’s disease, atherosclerosis, and aging. During aging, fibroblast–ECM interactions become disrupted due to the fragmentation of collagen fibrils, as with time, fibroblasts synthesize fewer ECM proteins and more matrix-degrading metalloproteinases. This imbalance of ECM homeostasis further drives collagen fibril fragmentation in a self-perpetuating cycle. Therefore, the following questions arise: Can cosmetics somehow support fibroblasts or ECM components against the damaging effects of MMPs, and can the ingredients of the cosmetics affect the ECM?

The literature on the theme of cosmetic preservatives is widely described. Nowadays, there is a strong tendency of using products with no paraben content [6]. To our knowledge, there are either no or only poorly developed scientific reports proving the effectiveness and safety of chosen substances on skin cells or their models. This article checked if these substances could serve as preservatives and if they could affect the dermal ECM and the mechanisms by which these changes alter the interplay between fibroblasts and their extracellular matrix.

## 2. Materials and Methods

### 2.1. Reagents

Diazolidinyl urea (germall II) and imidazolidinyl urea (germall 115) were obtained from Sigma Aldrich (St. Louis, MO, USA). Phenoxyethanol and GSB preservative were obtained from Zrób Sobie Krem (Prochowice, Poland). Methyl paraben and propyl paraben were products of Stanlab (Lublin, Poland). One of the tested grapefruit oils was from Profarm (Lębork, Poland)—Citrus Grandis (Grapefruit) Peel Oil; the second was a home-made oil distilled from fresh peel of grapefruit. All used preservatives were dissolved in dimethyl sulfoxide (DMSO) (Sigma Aldrich, St. Louis, MO, USA) to make 10 mg/mL stock solutions. The chemical structures of the used compounds are presented in Table 1.

### 2.2. In Vitro Antimicrobial Assay

The examined compounds were screened in vitro for antibacterial and antifungal activities using the broth microdilution method according to the European Committee on Antimicrobial Susceptibility Testing (EUCAST) [22] and Clinical and Laboratory Standards Institute (CLSI) guidelines [23] against a panel of reference and clinical or saprophytic strains of microorganisms, including Gram-positive bacteria (*Staphylococcus aureus* ATCC 25923, *Staphylococcus aureus* ATCC 43300, *Staphylococcus aureus* ATCC 6538, *Staphylococcus aureus* ATCC 29213, *Staphylococcus epidermidis* ATCC 12228, *Enterococcus faecalis* ATCC 29212, *Streptococcus pyogenes* ATCC 19615, *Bacillus subtilis* ATCC 6633, *Bacillus cereus* ATCC 10876, *Micrococcus luteus* ATCC 10240, and *Propionibacterium acnes* ATCC 11827), Gram-negative bacteria (*Bordetella bronchiseptica* ATCC 4617, *Escherichia coli* ATCC 25922, *Klebsiella pneumoniae* ATCC 13883, *Proteus mirabilis* ATCC 12453, *Salmonella* Typhimurium ATCC 14028, and *Pseudomonas aeruginosa* ATCC 9027), and fungi belonging to yeasts (*Candida albicans* ATCC 2091, *Candida albicans* ATCC 10231, *Candida parapsilosis* ATCC 22019, *Candida glabrata* ATCC 90030, and *Candida krusei* ATCC 14243) and mold (*Aspergillus niger* ATCC 16404). The microorganisms belonging to ATCC came from the American Type Culture Collection, routinely used for the evaluation of antimicrobials. All the used microbial cultures were first subcultured on nutrient agar or Sabouraud agar at 35 °C for 18–24 h or 30 °C for 24–48 h for bacteria and fungi, respectively.

Samples containing the examined compounds were dissolved in DMSO. Ciprofloxacin, vancomycin, or nystatin (Sigma Aldrich, St. Louis, MO, USA) were used as a reference antibacterial or antifungal compound. Subsequently, the MIC (minimal inhibitory concentration) of the substances was examined by the microdilution broth method, using their two-fold dilutions in Mueller–Hinton broth or Mueller–Hinton broth with sheep’s blood (for bacteria) and RPMI 1640 broth with MOPS (for fungi) prepared in 96-well polystyrene plates. Final concentrations ranged from 16,000 to 8 µg/mL for the grapefruit essential oils and from 4000 to 2 µg/mL for the other compounds. Microbial suspensions were prepared in 0.85% NaCl with an optical density of 0.5 McFarland standard. Next, each bacterial or fungal suspension was added to each well containing broth and various concentrations of the examined compounds. After incubation, the MIC was assessed spectrophotometrically as the lowest concentration of the samples showing complete bacterial or fungal growth inhibition. Appropriate DMSO, growth, and sterile control groups were included in the experiment. The medium with no tested substances was used as a control [24,25,26].

The MBC (minimal bactericidal concentration) and MFC (minimal fungicidal concentration) are defined as the lowest concentration of the compounds that is required to kill a particular bacterial or fungal species, respectively. MBC/MFC were determined by removing the culture using MIC determinations from each well and spotting onto the appropriate agar medium. The plates were incubated under appropriate conditions for bacteria and fungi. The lowest compound concentrations with no visible growth observed were assessed as bactericidal/fungicidal concentrations. The MBC/MIC and MFC/MIC ratios were calculated to determine bactericidal/fungicidal (MBC/MIC ≤ 4, MFC/MIC ≤ 4) and bacteriostatic/fungistatic (MBC/MIC > 4, MFC/MIC > 4) effects of the tested compounds. All the experiments were repeated three times, and representative data are presented [24,25,26].

### 2.3. Cell Viability Assay

Normal newborn fibroblasts (BJ; ATCC CRL-2522, American Tissue Type Collection, Manassas, VA, USA) were grown in Eagle’s Minimum Essential Medium (EMEM) supplemented with 10% fetal bovine serum (FBS), 2 mM L-glutamine, 100 U/mL penicillin, and 100 μg/mL streptomycin. They were purchased from Sigma Aldrich (St. Louis, MO, USA). Additionally, 1xMEM Nonessential Amino Acids and 1 mM sodium pyruvate from Corning (Manassas, VA, USA) were added to the medium. Cells used for tests were in early passages (3–10) and seeded until they reached 80–90% confluency. Human immortalized keratinocytes (HaCaT, CLS CVCL_0038, Cellosaurus Cell Lines, Geneva, Switzerland) were cultured in Dulbecco’s Modified Eagle Medium (DMEM) with the following additives: 10% FBS, 2 mM L-glutamine, 100 U/mL penicillin, and 100 μg/mL streptomycin, purchased from Sigma Aldrich (St. Louis, MO, USA). All cell lines were grown and maintained at 37 °C in 5% carbon dioxide.

To assess cell viability, the Promega MTS assay (Madison, WI, USA) was used. The test cells were cultured in a 96-well plate at 5000 cells/100 μL well (BJ) and 10,000 cells/100 μL well (HaCaT). After 24h, full medium (with FBS) was removed. The seeded cells were incubated with applicable solutions of preservatives used in cosmetic products for 24 h at various concentrations (0.78, 1.56, 3.125, 6.25, 12.5, 25, 50, and 100 µg/mL) prepared in serum-free medium (SFM). The concentrations of the tested substances were standardized through the solubility limits of the substances. To standardize the concentrations and maintain identical experimental conditions for each substance, optimal dilutions were chosen. DMSO concentration is an important factor limiting cell viability. Each solution was made to maintain 1% DMSO concentration in each tested and control well. After the incubation time, MTS reagent was added to each of the wells at a volume of 20 μL per well, and an additional 3 h incubation time was required. The results were measured by recording absorbance at 490 nm, for which the ELx800 Plate Reader from BioTek Instruments, Winooski, VT, was used. Each treatment was performed in six replications, and two to three independent experiments were carried out. Cell viability in the presence of the examined compounds was calculated as a percentage of the control cells [27]. The four preservatives which had the most notable activity were chosen for further experiments. Two types of grapefruit oils (freshly distilled and commercial) and two types of germalls (IU and DU) were examined to compare them.

### 2.4. Sircol Collagen Assay

Fibroblasts were seeded in full medium to 6-well plates at the same density of cells per area as in the 96-well plate—8300 cells/2 mL well. Cells were allowed to attach overnight. After that, the medium was replaced with SFM, and the cells were starved for 3 h. Stock solutions of concentrations of 100 mg/mL were made to obtain 0.1% DMSO in each tested and control well. BJ cells were exposed to 100, 25, and 5 µg/mL of germalls, grapefruit oils, and vehicle solutions for 24 h. One milliliter of the medium from each well was collected in Eppendorf tubes to make samples for measuring total soluble collagen using the Sircol collagen assay kit from Biocolor (Carrickfergus, UK). The medium was centrifugated to dispose of the detached cells caused by treatment, especially in the highest concentrations of preservatives. Then, 200 μL Isolation and Concentration Reagent was added to the vials and incubated at 4 °C overnight. Concentrated collagen was made dense by centrifugation. Next, 1 mL of Sircol dye was added to 100 μL tubes content and incubated for 30 min on a gentle orbital shaker. Following this, the next centrifugation step was made, and the visible collagen pellet was washed once and dissolved in an Alkali Reagent. The same steps were taken to obtain a calibration curve from the collagen standard. The absorbance of the samples was measured at 562 nm using a reader (ELx800 Plate Reader BioTek Instruments, Winooski, VT, USA). Experiments were performed in six independent replicates [28].

Every well from the treatment dish was visualized by a Leica DMi1 microscope with a camera from Leica Microsystems, Wetzlar, Germany.

### 2.5. Western Blot Analysis

Fibroblast cells plated and treated for the Sircol test were used to measure changes in collagen III type secretion in the cells. Cell surfaces were rinsed by cold Dulbecco’s phosphate-buffered saline (DPBS) without calcium and magnesium (Sigma Aldrich). The plates were placed on ice, and a protease and phosphatase inhibitor cocktail from Thermo Fisher (Waltham, MA, USA) was diluted in cell lysis buffer, purchased from CellSignaling (Danvers, MA, USA), to induce cell lysis. Aliquots of the cell lysate were used for protein determination using the BCA Protein Assay kit QPRO Cyanagen (Bologna, Italy). Equal amounts of protein (average about 2µg per well) from the lysates were separated by 4–12% sodium dodecyl sulfate (SDS) polyacrylamide gel (Thermo Fisher) electrophoresis under reducing conditions and electrotransferred onto nitrocellulose membranes using wet transfer (Bio-Rad, Hercules, CA, USA). Every gel after the transfer was checked using Coomassie brilliant blue staining (Sigma Aldrich, St. Louis, MO, USA). Membranes were blocked in 5% non-fat dry milk and incubated with primary antibodies overnight at 4 °C and then for 1 h with secondary antibody anti-rabbit conjugated with horseradish peroxidase enzyme (CellSignaling, Cat# 7074). Primary recombinant monoclonal rabbit antibody was purchased from Abcam (Cambridge, GB), raised against collagen III (Cat# ab184993, EPR 17673), 1:1000 diluted. The detection of bands of interest was performed by chemiluminescence using the Westar Supernova substrate (Cyanagen) and the Azure c400 imaging system by Azure Biosystems (Dublin, CA, USA). Protein bands were quantified by measuring densitometry using ImageJ software (National Institutes of Health, Bethesda, MD, USA). The membranes were reused for the detection of Hsp90 protein using a rabbit monoclonal anti-Hsp90 antibody, diluted 1:1000 (CellSignaling, Cat# C45G5). Detection of Hsp90 served as an equal protein loading control [29,30].

### 2.6. Zymographic Determination of MMP-2 Activity

The activity of MMP-2 was determined through gelatin zymography based on the visualization of free-gelatin areas digested by MMP-2. Briefly, medium samples were mixed with sample buffer containing 10% SDS in a 1:4 proportion. Electrophoresis of the samples on 10% gel polyacrylamide gel with 0.05% gelatin type A from porcine skin (G2500) (Sigma–Aldrich, USA) was performed to separate the enzyme. The gels were washed for 1 h to remove the SDS and then incubated overnight at 37 °C in a buffer containing 1% Triton X-100 (pH 7.2). Gels were stained using 0.1% Coomassie Blue R-250 in 20% methanol and 10% acetic acid, and subsequently, the stain was removed in 20% methanol and 10% acetic acid. MMP-2 activity was detected as clear bands on a blue background. Enzyme was identified by comparing its localization with standards of gelatinases (R&D Systems Inc., Minneapolis, USA). Zymographic gels were scanned and quantified using ImageJ software (National Institute of Health, Bathesda, USA). The results were expressed as the % activity of secreted MMP-2 versus the non-treated control group (100% of activity) [31].

The protein concentration in the cell sample medium was determined using the bicinchoninic acid reagent (Thermo Fisher Scientific, Waltham, MA, USA).

### 2.7. Statistics

Experiments were replicated at least three times unless otherwise specified. Data are presented as mean ± standard deviation (SD). Error bars indicate the SD. The featured scores were subjected to statistical analysis using one-way ANOVA followed by Dunnett’s multiple comparison test (*, *p* < 0.05; **, *p* < 0.01; ***, *p* < 0.001) using GraphPad Prism 9 (Graph Pad Software Inc., San Diego, CA). This test was used to establish statistical significance between the mean of every preservative’s concentration to the vehicle control. Values of *p* < 0.05 were considered statistically significant.

## 3. Results

### 3.1. Antibacterial Activity

Our data, presented in Table 2 and Table 3, showed some antimicrobial activity of the tested natural and synthetic substances. Their minimal inhibitory concentrations (MICs) and minimal bactericidal concentrations (MBCs) ranged from 125 µg/mL to >16,000 µg/mL.

The compounds germall II and germall 115 exhibited the highest activity towards bacteria (MIC and MBC = 125–1000 µg/mL). Streptococcus pyogenes ATCC 19615, Micrococcus luteus ATCC 10240, and Propionibacterium acnes ATCC 11827 (belonging to Gram-positive bacteria) were the most sensitive to germall II (MIC = 125 µg/mL, MBC = 125–500 µg/mL), while Bordetella bronchiseptica ATCC 4617 (from Gram-negative bacteria) was most sensitive to both germalls (MIC = 125 µg/mL and MBC = 250–500 µg/mL). These substances showed a bactericidal effect (MBC/MIC = 1–4) towards all reference bacteria. Parabens compounds indicated slightly lower activity against bacteria, with MIC = 250–2000 µg/mL and MBC = 250–>4000 µg/mL. Most Gram-positive bacteria were more susceptible to propyl paraben, which had mainly bactericidal activity, than to methyl paraben. The MIC of methyl paraben ranged from 1000 µg/mL to 2000 µg/mL, except with *B. bronchiseptica* ATCC 4617 (MIC = 250 µg/mL and MBC = 2000 µg/mL). Meanwhile, the minimal concentrations of phenoxyethanol and GSB preservatives which inhibited the growth of bacteria were 2000–≥4000 µg/mL, and the minimal concentrations which killed them were ≥4000 µg/mL. The bacteria most sensitive to phenoxyethanol and GSB were *M. luteus* ATCC 10240 and *B. bronchiseptica* ATCC 4617 (MIC = 2000 µg/mL). In the case of the two essential oils from grapefruit (commercial and freshly distilled), antimicrobial activity against reference bacteria was comparable. MIC and MBC values ranged from 8000 µg/mL to ≥16,000 µg/mL. The exceptions were two bacteria, *S. pyogenes* ATCC 19615 and *P. acnes* ATCC 11827, with MIC and MBC = 4000–8000 µg/mL and a bactericidal effect (MBC/MIC = 1–2).

### 3.2. Antifungal Activity

These results also indicated some antifungal effects of the tested substances (Table 4). Among them, the highest activity against the reference fungi was exhibited by both parabens, with MIC = 125–500 µg/mL, MFC = 250–2000 µg/mL, and a fungicidal effect (MFC/MIC = 1–4). Fungi belonging to yeasts, namely *Candida parapsilosis* ATCC 22019 and *Candida krusei* ATCC 14243, were the most sensitive to propyl paraben (MIC = 125 µg/mL, MFC = 250 µg/mL, and MFC/MIC = 2). In the case of both germalls and phenoxyethanol, activity varied slightly. The minimal concentrations of these compounds which inhibited the growth of fungi and killed them were 500–4000 µg/mL. Therefore, they indicated a fungicidal effect (MFC/MIC = 1–4). In turn, the activity of the GSB preservative was the weakest and was the same against all fungi (MIC and MFC = 4000 µg/mL and MFC/MIC = 1).

The antimicrobial effect of the grapefruit essential oils against fungi was similar. These oils exhibited slightly stronger activity against yeasts from *Candida* spp. (MIC = 2000–4000 µg/mL, MFC = 2000–16,000 µg/mL) than on mold *Aspergillus niger* ATCC 16404 (MIC = 8000 µg/mL, MFC = 8000–16,000 µg/mL, and MFC/MIC = 1–2).

### 3.3. Effect of Preservatives on Cell Viability

Cellular viability was measured using the MTS assay at different concentrations on fibroblasts and keratinocytes. As seen in Figure 1 and Figure 2, the type of preservative used in the cosmetic product had different effects on the cytotoxicity to skin cells and thus on cell viability. Depending on the used preservative, cell viability decreased to a greater or lesser value. Propyl and methyl parabens had a similar cytotoxic effect on both cell lines. The influence on viability was also similar to the results for the phenoxyethanol and GSB preservatives. Furthermore, the formaldehyde releasers (germalls II and 115) acted two ways. At a concentration of 10 µg/mL, cell viability decreased in both cases and in both cell lines. After increasing the concentrations of both preservatives, the viability increased, but later, that for germall II decreased to the lowest viability value. This led to the conclusion that at higher concentrations, germall II is more toxic than germall 115. The grapefruit oils were the most cytotoxic to cells compared to all cases considered. It should be noted that the distilled grapefruit oil was less cytotoxic for keratinocytes than the bought one (industrially produced). Compared to keratinocytes, fibroblasts are more sensitive to the effects of irritants, such as preservatives, which are discussed in this paper. This is due to their location in the layers of the skin and their function.

### 3.4. Influence of Preservatives on Collagen Synthesis

To check the influence of the preservatives on the skin, especially on collagen biosynthesis, which is important to users of skin beautifiers, we measured the amount of total collagen using the Sircol assay. Tests were provided only for four substances because in the literature, there is lack of information about this topic for the other preservatives. We found that there was no significant decrease in the production of total collagen by fibroblasts (Figure 3a) at this incubation time. After the incubation time, in commercial grapefruit oil, an increase in the amount of total collagen in medium, especially at 100 μg/mL, could be seen. Compared to photos (Table 5), these cells were disrupted, and it could lead to suspicions that collagen could be released by detached cells. The incubation time (24 h) of the fibroblast cells with the preservative could have been enough to synthesize the collagen by all metabolically active cells (until they were active), and then, through the cytotoxic effect of the studied compounds, it could have been released from inside by the disruption process. We suspect this is the reason that the amount of collagen present in the medium (Figure 3a) did not change in contrast to the amount of collagen inside the cells, as assessed using the Western blot technique (Figure 3c). It could cause false positive results despite the centrifugation to get rid of these cells. The structure of the BJ cells was nearly destroyed. They were peeled off the surface because of the strong influence of this potential preservative at this concentration.

In the case of the freshly distilled homemade grapefruit oil and germall 115, the improvement in the level of total collagen in medium was not statistically significant compared to the control group. The reduction in collagen level in germall II after treatment with a concentration of 25 μg/mL had significance in statistics only. Comparing this result to the photo, cells were not damaged at all (as in commercial grapefruit oil at 100 μg/mL). The principle of activity could be different. Presumably, it affected collagen biosynthesis not only for viability.

The effect on the level of intracellular collagen III is presented in Figure 3b,c. We observed that in this group, the germalls were the most destructive. The homemade grapefruit oil (distilled) was the mildest one. The post hoc Dunnett test indicated that this type of collagen significantly decreased only for the highest doses, as shown by the COL3 value observed for germall II and 115 (*p* < 0.001, Figure 3b). Moreover, germall II is safer than germall 115 considering the improvement in collagen synthesis at some concentrations.

It can be seen that intracellular production of collagen decreased more than in collagen present in medium.

### 3.5. MMP-2 Activity

Figure 4 presents the activity of MMP-2 secreted by fibroblasts (BJ cell line) stimulated by different doses of the preservatives. Statistical analysis proved a significant impact of the used compounds (germall II, germall 115, and commercial and freshly distilled grapefruit oils) on the activity of secreted MMP-2. MMP activity on collagen was the lowest using freshly distilled (homemade) grapefruit oil as a preservative. At the majority of concentrations, no statistically significant effect was observed compared to the control group. In other cases, the synthesis inhibition of collagen (Figure 3b,c) was related to the gradual growth of MMP (Figure 4).

## 4. Discussion

Our results indicate that the selected studied compounds (Table 2, Table 3 and Table 4) showed antimicrobial activity. Among them, germalls were particularly effective against both Gram-positive and Gram-negative bacteria. Meanwhile, GSB and propyl paraben were especially active against fungi belonging to *Candida* and *Aspergillus*. Proper functioning of the skin ECM determines its healthy appearance and firmness and delays the aging processes. Therefore, many cosmetic products contain substances that support the good condition of ECM in the skin. However, additional substances found in cosmetics, such as preservatives or fragrances, can also exert some effects on the skin and its components. Although safety tests are performed for all ingredients, their exact effects on ECM are not well known. Studies conducted on one paraben, namely methyl paraben, revealed increased activity of MMP-2 with decreased activity of TIMP-2. This could explain the methyl paraben-induced decrease in collagen concentration. It may result not only from the reduced synthesis of collagen but also from increased degradation through enhanced activation of MMP-2 [28]. Parabens, in connection with their bad reputation, were thoroughly examined. Due to their many advantageous properties, they are widely applied in cosmetics, food products, and pharmaceuticals, but recent research has uncovered their ability to accumulate in tissues and exert many adverse effects. Our research, for the first time, addressed the overall effect of other preservatives on skin ECM by studying their effect on MMP-2 activity. Except for cytotoxicity and contact sensitivity tests, there are no studies of their effect on skin ECM in the available literature.

As collagen is an essential main building block of the skin, providing its strength and elasticity, the degradation of collagen results in the wrinkling and sagging of the skin. Physiologically, in healthy, youthful skin, the synthesis and degradation of the ECM are balanced. Dermal fibroblasts not only produce and organize the ECM of the dermis but also generate active enzymes, such as MMPs, able to degrade the degrade the ECM components in normal state and disease. Human skin, like all other organs, undergoes chronological aging, but unlike other organs, it is in direct contact with environmental agents. Therefore, it is exposed to additional external factors such as UV radiation which may directly or indirectly affect the ECM and thus cause skin lesions and acceleration the aging process [32].

Systematic studies showed that several MMPs, including MMP-2, are capable of cleaving collagen. Moreover, greater degradation of type I collagen, i.e., the mature type, was connected with the detection of higher activity of the gelatinases MMP-2 and MMP-9 in skin [33]. We could conclude that increased MMP-2 activity could be responsible for greater connective tissue degradation and therefore premature skin aging. During senescence, the expression of MMPs increases, but the expression of its naturally occurring tissue inhibitors (TIMPs) decreases, making MMPs more active. If, additionally, any of the cosmetics components could induce higher MMP expression, the process could be accelerated.

This study shows that the type of preservative used in a cosmetic product influences the viability of skin cells differently. Propyl and methyl parabens were characterized by a similar cytotoxic effect in both cell lines. The effect on viability was similar to the test results for phenoxyethanol and GSB. The formaldehyde releasers (germall II and 115) presented in the study had different ways of influencing cell lines. At a concentration of 10 µg/mL, cell viability decreased in both cases and both cell lines. After increasing the concentration of both preservatives, the viability increased, but for germall II, it then decreased to the lowest value of viability. This led to the conclusion that at higher concentrations, germall II is more toxic than germall 115. Grapefruit oil was the most cytotoxic for cells in comparison to all analyzed cases.

Research by Sakaguchi et al. (2006) shows that methyl paraben is characterized by the lowest cytotoxicity compared to germall (II and 115) and formaldehyde against THP-1 cells. These cells are an acute monocytic leukemia line and were the model in the testing of substances considered to be skin sensitizers. Compared to the research conducted in this study, methyl paraben also turned out to be the least cytotoxic, and the results are also consistent in the comparison of germalls [4].

The team of Ishiwatari (2006) evaluated the permeability of methyl paraben through individual skin layers. The highest amount of the preservative accumulated mainly in the epidermis; it also penetrated to a small extent into the dermis and subcutaneous tissue. According to the results of in vitro experiments, methyl paraben was not completely metabolized and may have accumulated in the skin. To provide credibility to the in vitro data obtained, an in vivo study was conducted. After multiple applications of a product containing the tested paraben, the absorption in the stratum corneum was significant; after discontinuing the application of the product, it immediately decreased to a minimum value (before use). Ishiwatari et al. stated that long-term use of paraben-containing products affects the proliferation of keratinocytes (NHEKs—normal human epidermal keratinocytes). Only after about 40 days of use, cell proliferation was reduced to a small degree compared to the negative control. This case may explain the results obtained in the present study conducted on keratinocytes (HaCaT cell line), which illustrate a slight decrease in viability after 24 h of incubation with the selected paraben [34].

The paper published by Dubey et al. confirms previous findings on the impact of methyl paraben. It acknowledged that ultraviolet radiation affects the cytotoxicity of the paraben against the HaCaT cell line used in their study. From the MTT assay conducted in the study, it was concluded that exposure to UVB and sunlight in connection with methyl paraben causes phototoxicity. The study also confirmed the effect of the tested compound on fibroblasts by measuring fibroblast viability (PCS-201-012) against increasing concentrations of methyl paraben, showing that their mitochondrial activity decreases. Cytotoxicity was assessed using the MTT assay, which is similar to the MTS assay used in our study [35].

As a preservative, 2-phenoxyethanol is generally considered one of the milder ones, as confirmed by studies. However, there are reports of its allergy-inducing potential and possibly neurotoxic effects. Jung and Lee presented results after 24 h of incubation—a higher concentration of phenoxyethanol caused a significant reduction in HaCaT cells viability (the most after applying very high concentrations of 60 mM). The study also showed that the length of incubation time does not affect the cytotoxic effect [7].

The preservative GSB obtained similar cytotoxicity results to phenoxyethanol, which may suggest that the addition of the carbohydrate provides a gentler effect on skin cells. According to a patent registered in the U.S., gluconolactone in personal care cosmetics shows anti-irritant effects against other substances in the formulation. A reduction in skin inflammation was observed for gluconolactones measured against fibroblasts by Graves et al. by monitoring interleukin one (IL-1), the level of which decreased significantly (IL-1 is an inducer of their formation). GSB is a compound that consists of a sugar molecule and sodium benzoate, well known as a common food preservative. The sugar molecule in this case reduces the cytotoxic properties of sodium benzoate to skin cell lines. The studies of Alabsolghar et al. (2017) and H.-W. Park (2009) confirmed its non-cytotoxic character against human and rat cells. The safe, acceptable concentration according to both authors was 1 mg/mL [36,37,38,39].

More recently, the cosmetic manufacturing industry has begun a new trend of substituting chemical preservatives with only natural substances. Plants exhibit antibacterial and antifungal properties, and their effectiveness and efficiency are still being analyzed.

The possibility of using essential oils and plant extracts in cosmetics may be limited by their difficult solubility, characteristic scent, interaction with other components, as well as cytotoxicity [6].

Grapefruit oil is a plant substance that acts as an antifungal, and it was highly cytotoxic to both cell lines. To achieve the targeted effect and safe use in cosmetic formulations, it must be diluted appropriately. Heggers et al. performed in vitro experiments on human fibroblasts which demonstrated that dilutions of the oil above 1:256 preserve the viability of the cells. Formulation of a product with grapefruit oil at a ratio of 1:1 proved to be highly toxic, leading to the destruction of the structure of these cells [40].

The correct antifungal properties and low cytotoxicity of parabens proven in our studies show that parabens are safe and effective preservatives in cosmetics. Many misconceptions have arisen about the negative health effects of parabens. When Darbre et al. in 2004 published a study suggesting a link between the use of personal care products containing parabens and their concentration in breast cancer tissue, parabens began to be known as “cancer-causing preservatives”. Although the observations of Darbre et al. were never confirmed by other researchers, today, many cosmetics are labeled as “paraben free”. Moreover, the tag “paraben free” is not officially registered or required in the European Union [6,41].

We also determined the influence of germall II, germall 115, and commercial and freshly distilled homemade grapefruit oils on the activity of MMP-2 by the zymographic method. Our study showed a dose-dependent increase in the activity of secreted MMP-2 by cells treated with all examined substances up to the concentration of 25 µg/mL and then a decrease in activity at the concentration of 100 µg/mL. The highest rise in MMP activity was reported for germall II and 115, which was lower for marketed grapefruit oil, while for homemade grapefruit oil, the increase was not statistically significant. The decrease in metalloproteinase activity for the highest dose of the substances may have resulted from the decrease in cell numbers due to observed cytotoxic effects. According to the literature, the importance of MMPs is underestimated [21]. These enzymes matter because they can lead beneficial and destructive processes in the human body.

For our study, the most interesting role was the disruption of collagen. Most researchers are focused on parabens. There are few articles where the influence of this substance on skin cells and processes is presented. The study of Majewska et al. could serve as an example, where they wrote about methyl paraben. Activity of MMP-2 was induced as an increasing concentration was presented [28]. It was the reason that we investigated whether other, at present, widely used substances in cosmetics free of parabens (germalls) or promising new natural oils from grapefruit peel could serve as effective preservatives without the mentioned effect on MMP-2. Based on our results, we could conclude that freshly distilled homemade grapefruit oil should be the least destructive for skin ECM due to low MMP-2 activity in comparison with the not-treated control group and small influence on collagen III. According to the literature, the most commonly examined grapefruit origin product is GSE^®^ [40]. It is a known, industrially available preservative from grapefruit seed, but it is not of natural origin. This product contains some synthetic additives. The market trend is currently focused on nature, and cosmetics are mostly produced from natural ingredients. This is why the scientific world is working on using extracted substances from plants, e.g., essential oils (EOs). Using EOs in products is associated with various problems such as high volatility, low thermal and chemical stability, and low solubility in water. It could be difficult for manufacturers of natural cosmetics, but new research is focused on the encapsulation of selected EOs in nanocarriers [42]. This new technology could eliminate most of the difficulties and spread natural substances in everyday usage.

The findings of this study have to be seen while considering some limitations. The first is the viability test (MTS), which is only a preliminary test. Conducting proliferative marker measurement experiments could be beneficial in a broader assessment and would undoubtedly lead us to more complex conclusions. We would like to focus on the most notable compounds such as germalls. It could fill the literature gap we noticed in this topic. The second limitation concerns the amount of collagen produced. In future studies, we will aim to address whether collagen gene expression is regulated. Furthermore, our team conducted a series of experiments on a large group of compounds. In the future, we would like to perform more detailed research on selected compounds. This would allow us to determine which of them has the most promising application.

## 5. Conclusions

In conclusion, fibroblasts are more sensitive to the tested group of preservatives than keratinocytes. This could be caused by the differences between the cells’ structures. Keratinocytes deposit a structural protein, keratin, which can provide an additional barrier to protect the cell from external factors.

In the case of germalls, there was no correlation between the cell viability results and concentration. These compounds belong to the formaldehyde donor group. They exhibit antimicrobial activity due to the gradual release of methanal. These preservatives used at a concentration of 0.1% release 0.01% formaldehyde. Similar abnormal cytotoxic behavior has been reported in the literature [29]. We suspect that as formaldehyde donors, they may release it in a non-linear way. Formaldehyde is assessed as highly toxic and is banned in cosmetic products. That is why cosmetic industries find a way to use formaldehyde not as a single substances but combined. Germalls are characterized by a wide spectrum of activity at relatively low concentrations. Due to their high antimicrobial activity, it is worthwhile to carry out further studies of germalls. The selection of appropriate physical and chemical conditions for the cosmetic formulation could affect the replicability of the cytotoxicity results.

Essential oils, as substances of natural origin, are more likely to be used as ingredients in cosmetic formulas than synthetic ones. This is due to consumers’ perception of plant substances as being safer and environmentally friendly. In this article, we draw attention to this paradox because grapefruit oils are less effective against bacteria and fungi than methyl paraben. As the graphs indicate, the cytotoxicity of grapefruit oil is significantly higher compared to methyl paraben at the same concentrations. IC_50_ values were determined only in the case of grapefruit oils. When comparing essential oils with synthetic germalls, it should be noted that the use of natural derivatives in cosmetics also requires controlled doses.

## Figures and Tables

**Figure 1 cells-12-01076-f001:**
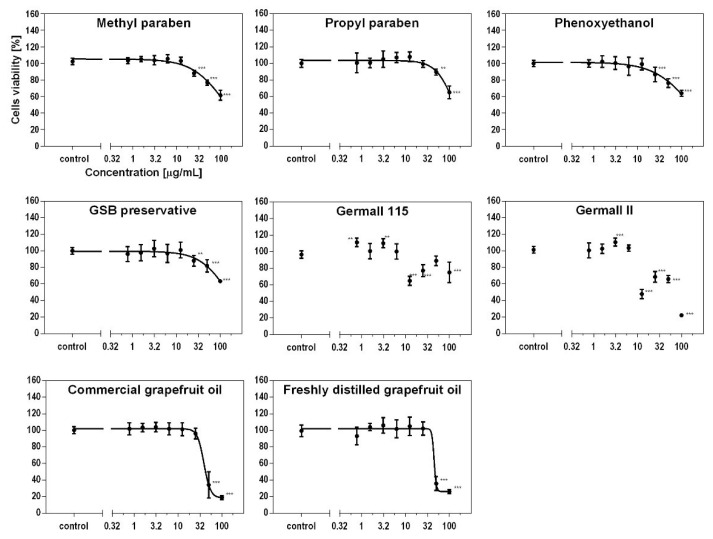
Effect of preservatives on fibroblast (BJ) cells’ viability using MTS test. Each value (mean ± SD) represents the average of cell viability (%) after 24 h incubation; n = 6, one-way ANOVA. **, *p* < 0.01; ***, *p* < 0.001.

**Figure 2 cells-12-01076-f002:**
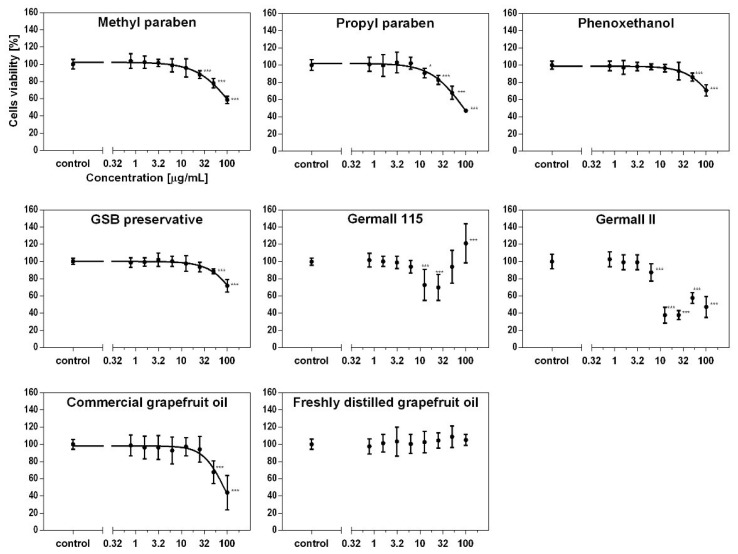
Effect of preservatives on keratinocyte (HaCaT) cells’ viability using MTS test. Each value (mean ± SD) represents the average of cell viability (%) after 24 h incubation; n = 6, one-way ANOVA. *, *p* < 0.05; ***, *p* < 0.001.

**Figure 3 cells-12-01076-f003:**
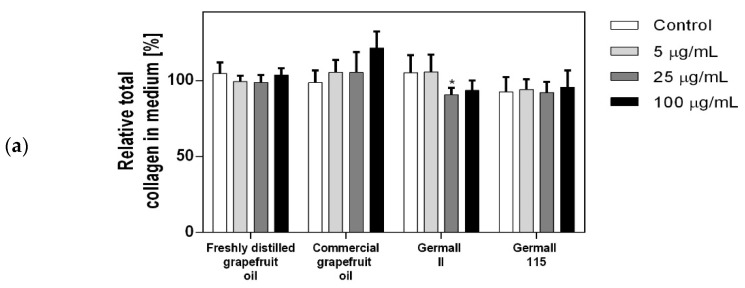
(**a**) Effect of selected preservatives on total content of secreted collagen in BJ cell medium using Sircol dye test. (**b**) Influence of the following compounds on intracellular collagen type III level, assayed by Western blot. (**c**) Level of collagen in BJ cells in each probe on membrane. (**d**) The same membrane as (**c**) but sharpened to visualize the lowest levels of collagen in germall probes. (**e**) Level of Hsp90 used as a control in each probe. Each value (% of control) ± SD represents the result after 24 h incubation of BJ cells with preservatives, repeated 6 times in independent experiments; n = 6, one-way ANOVA. *, *p* < 0.05; **, *p* < 0.01; ***, *p* < 0.001.

**Figure 4 cells-12-01076-f004:**
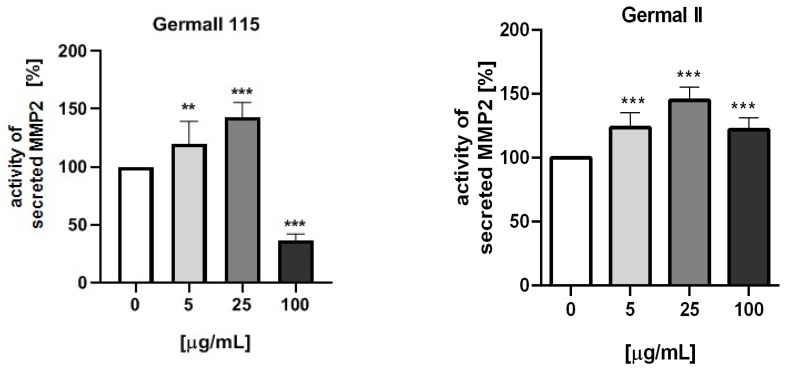
Activity of secreted MMP-2 by fibroblasts (BJ) stimulated by different doses of the most interesting preservatives. Values represent the mean (% of control) ± SD; n = 3, one-way ANOVA. *, *p* < 0.05; **, *p* < 0.01; ***, *p* < 0.001.

**Table 1 cells-12-01076-t001:** Structures of used compounds in the study.

Compound Name	Chemical Structure
Phenoxyethanol	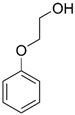
Germall II Diazolidinyl urea 1-(1,3-Bis(hydroxymethyl)-2,5-dioxoimidazolidin-4-yl)-1,3-bis(hydroxymethyl)urea	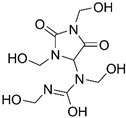
Germall 115 Imidazolidinyl urea 1,1′-Methylenebis(3-(3-(hydroxymethyl)-2,5-dioxoimidazolidin-4-yl)urea)	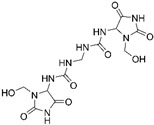
GSB Gluconolactone (70–80%) and sodium benzoate (22–28%)	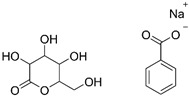
Methyl paraben	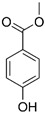
Propyl paraben	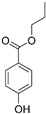
Grapefruit oil Citrus grandis oil	Undetermined composition. The main substances are flavonoids and ascorbic acid.

**Table 2 cells-12-01076-t002:** The activity data of tested compounds expressed as MIC (µg/mL), MBC (µg/mL), and MBC/MIC ratio against the reference strains of Gram-positive bacteria.

Species	MIC (MBC) and {MBC/MIC Ratio} of Compounds against Gram-Positive Bacteria								
	Commercial Grapefruit Oil	Freshly Made Grapefruit Oil	Germall II	Germall 115	Phenoxy-Ethanol	GSB Preservative	Propyl Paraben	Methyl Paraben	CIP/ VA
*Staphylococcus aureus* ATCC 25923	16,000 (>16,000) {>1}	8000 (>16,000) {>2}	**250** **(500)** **{2}**	**250** **(1000)** **{4}**	>4000 (>4000) {>1}	4000 (>4000) {>1}	**500** **(4000)** **{8}**	2000 (>4000) {>2}	0.48 (0.48) {1}
*Staphylococcus aureus* ATCC 6538	16,000 (>16,000) {>1}	8000 (>16,000) {>2}	**250** **(500)** **{2}**	**250** **(1000)** **{4}**	4000 (>4000) {>1}	4000 (>4000) {>1}	2000 (2000) {1}	2000 (>4000) {>2}	0.24 (0.24) {1}
*Staphylococcus aureus* ATCC 43300	16,000 (>16,000) {>1}	16,000 (>16,000) {>1}	**250** **(500)** **{2}**	**250** **(500)** **{2}**	>4000 (>4000) {>1}	4000 (>4000) {>1}	2000 (4000) {2}	2000 (>4000) {>2}	0.24 (0.24) {1}
*Staphylococcus aureus* ATCC 29213	16,000 (>16,000) {>1}	16,000 (>16,000) {>1}	**250** **(1000)** **{4}**	**500** **(1000)** **{2}**	>4000 (>4000) {>1}	4000 (>4000) {>1}	**1000** **(4000)** **{4}**	2000 (>4000) {>2}	0.48 (0.48) {1}
*Staphylococcus epidermidis* ATCC 12228	16,000 (>16,000) {>1}	16,000 (>16,000) {>1}	**250** **(500)** **{2}**	**500** **(1000)** **{2}**	>4000 (>4000) {>1}	4000 (>4000) {>1}	**500** **(2000)** **{4}**	2000 (2000) {1}	0.12 (0.12) {1}
*Enterococcus faecalis* ATCC 29212	16,000 (>16,000) {>1}	16,000 (>16,000) {>1}	**250** **(1000)** **{4}**	**500** **(1000)** **{2}**	>4000 (>4000) {>1}	4000 (>4000) {>1}	**1000** **(2000)** **{2}**	**1000** **(>4000)** **{>4}**	0.98 (1.95) {2}
*Streptococcus pyogenes* ATCC 19615	8000 (8000) {1}	4000 (8000) {2}	**125** **(125)** **{1}**	**250** **(250)** **{1}**	4000 (4000) {1}	4000 (>4000) {>1}	**250** **(250)** **{1}**	2000 (4000) {2}	0.24 (0.24) {1}
*Micrococcus luteus* ATCC 10240	8000 (16,000) {2}	8000 (>16,000) {>2}	**125** **(500)** **{2}**	**250** **(500)** **{2}**	2000 (4000) {2}	2000 (>4000) {>2}	**500** **(2000)** **{4}**	**1000** **(>4000)** **{>4}**	0.98 (1.95) {2}
*Bacillus subtilis* ATCC 6633	8000 (>16,000) {>2}	8000 (>16,000) {>2}	**250** **(500)** **{2}**	**500** **(500)** **{1}**	4000 (>4000) {>1}	4000 (4000) {1}	**500** **(2000)** **{4}**	2000 (4000) {2}	0.03 (0.03) {1}
*Bacillus cereus* ATCC 10876	16,000, (>16,000) {>1}	8000 (>16,000) {>2}	**250** **(1000)** **{4}**	**500** **(2000)** **{4}**	>4000 (>4000) {>1}	4000 (>4000) {>1}	**500** **(4000)** **{8}**	2000 (>4000) {>2}	0.06 (0.12) {2}
*Propionibacterium acnes* ATCC 11827	8000 (8000) {1}	4000 (4000) {1}	**125** **(125)** **{1}**	**250** **(250)** **{1}**	>4000 (>4000) {>1}	4000 (>4000) {>1}	**500** **(1000)** **{1}**	2000 (4000) {2}	–

Notes: Individual data of MIC, MBC, and MBC/MIC ratio are shown as MIC (without parentheses) and (MBC) and {MBC/MIC} (in special brackets), respectively. The standard antibacterial drugs ciprofloxacin (CIP—for all bacteria except *E. faecalis* ATCC 29212 and *S. pyogenes* ATCC 19615) and vancomycin (VA—for *E. faecalis* ATCC 29212 and *S. pyogenes* ATCC 19615) were used as positive controls; “–”, not tested.

**Table 3 cells-12-01076-t003:** The activity data of tested compounds expressed as MIC (µg/mL), MBC (µg/mL), and MBC/MIC ratio against the reference strains of Gram-negative bacteria.

Species	MIC (MBC) and {MBC/MIC Ratio} of Compounds against Gram-Negative Bacteria								
	Commercial Grapefruit Oil	Freshly Made Grapefruit Oil	Germall II	Germall 115	Phenoxy-Ethanol	GSB Preservative	Propyl Paraben	Methyl Paraben	CIP
*Bordetella bronchiseptica* ATCC 4617	16,000, (16,000) {1}	16,000 (16,000) {1}	**125** **(250)** **{2}**	**125** **(500)** **{4}**	2000 (4000) {2}	2000 (4000) {2}	**500** **(1000)** **{2}**	**250** **(2000)** **{8}**	0.98 (0.98) {1}
*Klebsiella pneumoniae* ATCC 13883	16,000, (>16,000) {>1}	16,000 (>16,000) {>1}	**250** **(500)** **{2}**	**500** **(500)** **{1}**	4000 (>4000) {>1}	4000 (>4000) {>1}	**1000** **(2000)** **{2}**	**1000** **(4000)** **{4}**	0.12 (0.24) {2}
*Proteus mirabilis* ATCC 12453	16,000, (>16,000) {>1}	16,000 (>16,000) {>1}	**500** **(500)** **{1}**	**250** **(500)** **{2}**	4000 (>4000) {>1}	4000 (>4000) {>1}	2000 (2000) {1}	**1000** **(4000)** **{4}**	0.03 (0.03) {1)
*Salmonella typhimurium* ATCC 14028	16,000 (>16,000) {>1}	16,000, (>16,000) {>1}	**250** **(500)** **{2}**	**500** **(1000)** **{2}**	4000 (>4000) {>1}	4000 (>4000) {>1}	2000 (4000) {2}	**1000** **(4000)** **{4}**	0.06 (0.06) {1}
*Escherichia coli* ATCC 25922	16,000 (>16,000) {>1}	16,000 (>16,000) {>1}	**250** **(500)** **{2}**	**500** **(500)** **{1}**	4000 (>4000) {>1}	4000 (>4000) {>1}	2000 (2000) {1}	**1000** **(4000)** **{4}**	0.004 (0.008) {2}
*Pseudomonas aeruginosa* ATCC 9027	16,000 (>16,000) {>1}	16,000 (>16,000) {>1}	**500** **(500)** **{1}**	**1000** **(1000)** **{1}**	4000 (>4000) {>1}	4000 (>4000) {>1}	2000 (>4000) {>1}	2000 (2000) {1}	0.48 (0.98) {2}

Notes: Individual data of MIC, MBC, and MBC/MIC ratio are shown as MIC (without parentheses) and (MBC) and {MBC/MIC} (in special brackets), respectively. The standard antibiotic ciprofloxacin (CIP) was used as a positive control.

**Table 4 cells-12-01076-t004:** The activity data of tested compounds expressed as MIC (µg/mL), MFC (µg/mL), and MFC/MIC ratio against the reference strains of fungi.

Species	MIC (MFC) and {MFC/MIC Ratio} of Compounds against Fungi								
	Commercial Grapefruit Oil	Freshly Made Grapefruit Oil	Germall II	Germall 115	Phenoxy-Ethanol	GSB Preservative	Propyl Paraben	Methyl Paraben	NY
*Candida albicans* ATCC 2091	4000 (8000) {2}	2000 (16,000) {8}	**1000** **(1000)** **{1}**	**1000** **(2000)** **{2}**	2000 (4000) {2}	4000 (4000) {1}	**250** **(500)** **{2}**	**500** **(1000)** **{2}**	0.24 (0.24) {1}
*Candida albicans* ATCC 10231	2000 (8000) {4}	2000 (16,000) {8}	**1000** **(2000)** **{2}**	2000 (4000) {2}	2000 (4000) {2}	4000 (4000) {1}	**250** **(500)** **{2}**	**500** **(2000)** **{4}**	0.48 (0.48) {1}
*Candida parapsilosis* ATCC 22019	2000 (16,000) {8}	2000 (16,000) {8}	**500** **(500)** **{1}**	**500** **(500)** **{1}**	1000 (4000) {4}	4000 (4000) {1}	**125** **(250)** **{2}**	**250** **(500)** **{2}**	0.24 (0.48) {2}
*Candida glabrata* ATCC 90030	4000 (16,000) {4}	4000 (16,000) {4}	**1000** **(1000)** **{1}**	**1000** **(1000)** **{1}**	4000 (4000) {1}	4000 (4000) {1}	**250** **(500)** **{2}**	**500** **(2000)** **{4}**	0.24 (0.48) {2}
*Candida krusei* ATCC 14243	2000 (2000) {1}	2000 (2000) {1}	**500** **(1000)** **{2}**	**1000** **(1000)** **{1}**	4000 (4000) {1}	4000 (4000) {1}	**125** **(250)** **{2}**	**500** **(2000)** **{4}**	0.24 (0.24) {1}
*Aspergillus niger* ATCC 16404	8000 (8000) {1}	8000 (16,000) {2}	**500** **(500)** **{1}**	2000 (4000) {2}	2000 (4000) {2}	4000 (4000) {1}	**250** **(500)** **{2}**	**500** **(1000)** **{2}**	–

Notes: Individual data of MIC, MFC, and MFC/MIC ratio are shown as MIC (without parentheses) and (MFC) and {MFC/MIC} (in special brackets), respectively. The standard antibiotic nystatin (NY) was used as a positive control; “–”, not tested.

**Table 5 cells-12-01076-t005:** Images of BJ cells after 24 h treatment with selected preservatives. Influence of the highest concentration on viability is observed as changes in structure of BJ cells. Magnification 100×.

	Control	5 µg/mL	25 µg/mL	100 µg/mL
Germall 115	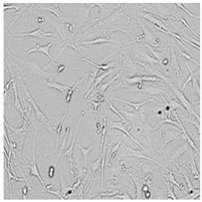	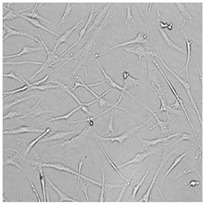	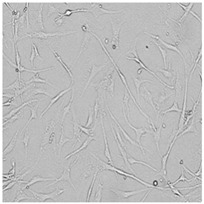	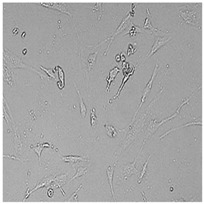
Germall II	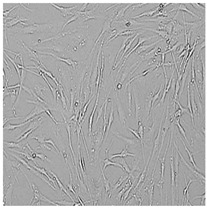	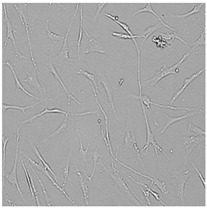	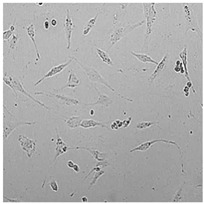	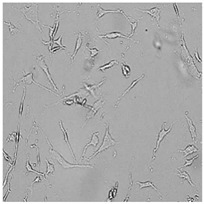
Commercial grapefruit oil	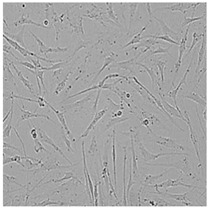	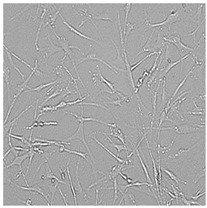	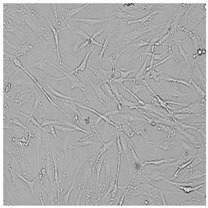	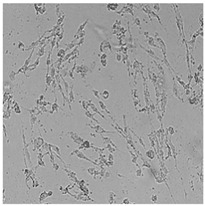
Distilled grapefruit oil	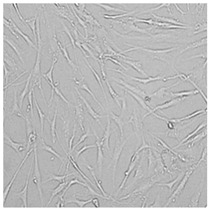	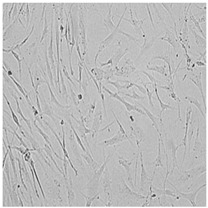	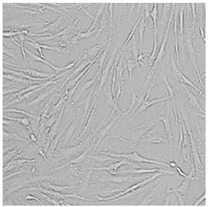	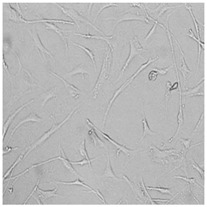

## Data Availability

Data are contained within the article.

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
