# Peer review of "Effect of Commonly Used Cosmetic Preservatives on Healthy Human Skin Cells"

_cells, 2023, doi:10.3390/cells12071076_

Round 1
Reviewer 1 Report (New Reviewer)
GÅ‚az P and colleagues analyzed in BJ cells the effects of various types of preservatives present in skincare products applied on the skin. They found the induction of MMP-2 activity at low concentrations. I have the following comments and suggestions regarding the content of manuscript.
-The abstract section is very general, it must contain the data related with relevant results and conclusions, respectively.
-In the introduction section, there are several phrases with only one reference, it needs a reference according to information in each phrase.
-Please check the acronyms, it is only necessary to define once.
-The authors should check if the reference 16 and 17 cover all information in the following phrase: “Although their substrates differ, all of them influence collagen degradation, i.e., collagenases could……………………………… type I, II, III, IV, VII, IX, X, XI collagen, gelatin, aggrecan, elastin, fibronectin, IGFBP, fibrine, fibrinogen, pro-MMP-9, pro-MMP-13, plasminogen, pro-TNFα [16,17].”
--There are several data required about cell culture. Please add description about cell culture, e.g., BJ cells number passage, the confluence percentage of seeded BJ cells; please detail the treatments of cells with the compounds, were cells seeded and treated at the same time? Did the culture media contain 10% fetal bovine serum when the treatments were performed?
-The reference 24 is not correct in the next phrase, please check it: “Cell viability in presence of examined compounds was calculated as a percentage of the control cells [24].”
-Authors should add the amount of total protein loaded in the western blot method. Why is Hsp90 protein used as equal protein loading control?
-Data from table 1, 2, and 3 are difficult to understand, please define data with and without parenthesis, and the other symbol used. What does mean the asterisk symbol?
-Figure 3 should contain the name of the cells where the experiments were performed. It must be used the same nomenclature in panel a) and b), compounds name.
-The authors should check the following sentences because they are confusing: ““We have found that there is no significant decrease in production of total collagen by fibroblasts (Figure 3A)”. However, the authors said the following: “Compared to photos (Table 2) these cells were disrupted, and it could lead to suspicions that collagen could be released by “dead” and detached cells.” How it could be shown that dead and detached cells can release collagen. In addition, the figure 3 (panel a), there is no significant decrease in production of total collagen by fibroblasts.
-Why were the compounds analyzed at 100 μg/mL if they have toxic effects on cell viability?
-I suggest to evaluate proliferative markers for those compounds that increase cell viability.
-I suggest to assess TIMP-2 expression by western blot to support the effects found on MMP2 activity.
-In the next phrase: “This could explain the MP-induced decrease in the collagen concentration, that may result not only from the reduced synthesis but also increased degradation through enhanced activation MMP-2 [20].” The reference is not correct.
-In discussion section, it should contain more information about the activity of MMP2 on collagen and the consequences for the skin.
-The authors should add the limitations of this study.
Author Response
Please find enclosed the revised manuscript cells-2279041 entitled “Effect of commonly used cosmetic preservatives on healthy human skin cells” by Patrycja GÅ‚az, Agata RosiÅ„ska, Sylwia Woźniak, Anna Boguszewska-Czubara, Anna Biernasiuk and Dariusz Matosiuk. We changed the manuscript substantially as suggested by the Reviewers.
Below are our answers:
Reviewer 1
GÅ‚az P and colleagues analyzed in BJ cells the effects of various types of preservatives present in skincare products applied on the skin. They found the induction of MMP-2 activity at low concentrations. I have the following comments and suggestions regarding the content of manuscript.
- The abstract section is very general, it must contain the data related with relevant results and conclusions, respectively.
Thank you for your comment. We added relevant results and conclusions according to your instructions.
- In the introduction section, there are several phrases with only one reference, it needs a reference according to information in each phrase.
Thank you for your valuable remark. The introduction was checked again, and missing references were added. We are using Mendeley Desktop to create references; we checked the references again and everything seems to be correct.
- Please check the acronyms, it is only necessary to define once.
Thank you for your comment, you are right. We checked the manuscript and delated double-explained acronyms or replaced full names with abbreviations where necessary.
- The authors should check if the reference 16 and 17 cover all information in the following phrase: “Although their substrates differ, all of them influence collagen degradation, i.e., collagenases could……………………………… type I, II, III, IV, VII, IX, X, XI collagen, gelatin, aggrecan, elastin, fibronectin, IGFBP, fibrine, fibrinogen, pro-MMP-9, pro-MMP-13, plasminogen, pro-TNFα [16,17].”
Thank you for your comment. We checked the manuscript and references again. The information was checked and replaced to correct data.
- There are several data required about cell culture. Please add description about cell culture, e.g., BJ cells number passage, the confluence percentage of seeded BJ cells; please detail the treatments of cells with the compounds, were cells seeded and treated at the same time? Did the culture media contain 10% fetal bovine serum when the treatments were performed?
Thank you for your comment. We added the descriptions with the number of BJ passages and their confluence percentage. We changed the treatment procedure; it should be more precise and better to understand. The cells were seeded first and then treated after 24 hours (to allow them to attach to the plate surface). The treatments of compounds were added. The text contains the information about the culture media – “The seeded cells were incubated with applicable solutions of preservatives used in cosmetic products for 24 h in various concentrations prepared in serum-free medium (SFM).”
- The reference 24 is not correct in the next phrase, please check it: “Cell viability in presence of examined compounds was calculated as a percentage of the control cells [24].”
Thank you for your remark. According to us, the reference is used correctly. In the supplementary material of cited publication is a procedure for the MTS test named as a “Cytotoxicity assessment” which we used as a model to make our procedure.
- Authors should add the amount of total protein loaded in the western blot method. Why is Hsp90 protein used as equal protein loading control?
Thank you for your valuable remark. Heat shock protein 90 (Hsp90) is one of the most abundant and ubiquitously expressed chaperone proteins, constituting approximately 1–2% of the total cell protein complement [reference: doi: 10.1371/journal.pone.0086842]. This protein is used in laboratory work as a loading control quite often as well as actin or tubulin. Our protein band of interest was about 250kDa (the band from Hsp90 is at 90kDa) and there was no suspicion that the amount of it will change in these conditions which is why we used that protein as a loading control. The amount of total protein which was loaded was added to manuscript.
- Data from table 1, 2, and 3 are difficult to understand, please define data with and without parenthesis, and the other symbol used. What does mean the asterisk symbol?
Thank you for your valuable comment. We fully agree with this suggestion. Therefore, we have included appropriate explanations below the Tables 2 – 4: “Individual data identified as MIC, MBC, and MBC/MIC ratio are shown without parentheses – MIC, and in special bracket – (MBC) and {MBC/MIC}, respectively.” and „Individual data identified as MIC, MFC, and MFC/MIC are shown without parentheses – MIC, and in special bracket – (MFC) and {MFC/MIC}, respectively.” We hope these explanations will be understandable to the reader. Moreover, asterisk symbol was used unnecessarily.
- Figure 3 should contain the name of the cells where the experiments were performed. It must be used the same nomenclature in panel a) and b), compounds name.
Thank you for your comment. The name of cell line was added in a description of the Figure 3 and the nomenclature was changed.
- The authors should check the following sentences because they are confusing: “We have found that there is no significant decrease in production of total collagen by fibroblasts (Figure 3A)”. However, the authors said the following: “Compared to photos (Table 2) these cells were disrupted, and it could lead to suspicions that collagen could be released by “dead” and detached cells.” How it could be shown that dead and detached cells can release collagen. In addition, the figure 3 (panel a), there is no significant decrease in production of total collagen by fibroblasts.
Thank you for your comment. You are right. It could be quite confusing. In first sentence which you remarked it should be added that according to results from Sircol assay (Figure 3A) there is no significant change in collagen observed in medium. The incubation time (24h) of fibroblasts cells with preservatives could be enough to synthesize the collagen by all metabolically active cells (until they were active) and then through cytotoxic effect of studied compounds could be released from inside of it by disruption process. We suspect this is the reason that amount of collagen presented in medium (Figure 3a) did not change in contrast to amount of collagen inside the cells – Western Blot technique (Figure 3c).
- Why were the compounds analyzed at 100 μg/mL if they have toxic effects on cell viability?
Thank you for your comment. Yes, we agree with you, but the cytotoxic effect is not notable in every compared compound. For example, germall 115 is not as toxic as other compounds (100ug/mL cytotoxicity is about 80%). To obtain comparable results we decided to check all of it in this particular concentration.
- I suggest to evaluate proliferative markers for those compounds that increase cell viability.
Thank you for your suggestion. Test of proliferative markers could be helpful in a wider evaluation and would certainly lead us to more complex conclusions. This is a very helpful suggestion for our further work to concentrate on this topic. Our team performed a variety of experiments on a large group of compounds. In the future, we would like to be able to perform more specific studies on selected compounds. It would allow us to evaluate the mechanisms of action of preservatives, and to find which of them have the most prospective use.
- I suggest to assess TIMP-2 expression by western blot to support the effects found on MMP2 activity.
In this study we wanted to reveal how the presence of preservatives from cosmetics influence on the activity of MMP-2 and then on the process of collagen degradation. TIMPs are considered as main tissue regulators of MMPs, however the regulation of MMPs activity is a complex process. As MMPs are able to degrade all ECM proteins, the process must be tightly controlled to prevent tissue destruction.
There are several steps to regulate activity of MMPs: transcription of the gene, activation of the proenzyme by removing the pro-peptide domain, the binding of natural inhibitors, pericellular or intracellular compartmentalization, allosteric activation, or oxidative modification. Finally, after the secretion from producing cells, MMPs activity can be regulated by inhibitors that could be categorizes in two groups: first, macromolecular, endogenous, natural such as TIMPs, and the second group of small molecules including exogenous natural or synthetic inhibitors. Among all these MMPs inhibitors, TIMPs have been found the most potent and specific physiological inhibitors [reference: doi: 10.2174/0929867325666180514111500.]
Therefore, determination of TIMP-2, will give us the information about regulation process occurring in skin, however, the result of gelatin zymography show us the final activity of MMP-2, what was the aim of our experiment.
- In the next phrase: “This could explain the MP-induced decrease in the collagen concentration, that may result not only from the reduced synthesis but also increased degradation through enhanced activation MMP-2 [20].” The reference is not correct.
Thank you for your comment. We changed the reference. It was not correct. We changed it for the right one.
- In discussion section, it should contain more information about the activity of MMP2 on collagen and the consequences for the skin.
Thank you for comment, we added more information about the activity of MMP2 on collagen and the consequences for the skin. Essential references were added.
-The authors should add the limitations of this study.
Thank you for your comment. We are aware of the limitations of our study. This topic is very interesting and perspective. We are planning evaluation studies of action mechanisms preservatives which will be public in the following study.
Revisions to the manuscript were marked with the "Track Changes" function. The manuscript has undergone English revisions.
We would like to thank the Reviewers for their time and all their comments. We would greatly appreciate it if you could consider publishing the manuscript with relevant corrections and additions.
We wish to join these following statements:
We confirm that neither the manuscript nor any parts of its content are currently under consideration or published in another journal.
All authors have approved the manuscript and agree with its submission to Cells.
We look forward to hearing your comments and decision concerning the publication of our manuscript.
Best regards
Patrycja GÅ‚az & Dariusz Matosiuk
Corresponding author:
Dariusz Matosiuk
Medical University of Lublin, Chair and Department of Synthesis and Chemical Technology of Pharmaceutical Substances, 4A Chodzki Str., 20-093 Lublin, Poland.
E- mail: dariusz.matosiuk@umlub.pl
Tel: +48 81 448 72 72
Reviewer 2 Report (New Reviewer)
Today, the use of cosmetic products accompanies our lives from the cradle to the grave. It is therefore highly desirable to have a thorough knowledge of all their chemical, physicochemical and toxicological (cytotoxicity) properties. The manuscript is very useful in this respect. Parabens are often used in small amounts as preservatives in cosmetics, pharmaceuticals, foods, and beverages since the 1920s. Researchers have identified parabens as chemicals that can potentially disrupt hormonal systems in both males and females. Some patients were found to have a sensitivity, allergic reactions to parabens. The European Union banned isopropylparaben and isobutylparaben from use in any personal care products in 2015 until more research can be done. Diazolidinyl urea (please, add its newly determined structure) and its chemically related to imidazolidinyl urea acts as a formaldehyde releaser. Both are skin allergens.
The article is well written, however I have a few comments. I do not recommend the use of any abbreviations in the abstract, which are not explained (ECM, or GSB and MMP-2). The text should consistently spell out Methyl paraben vs. methyl paraben (p. 14 etc.). A reader with a basic science background would welcome the chemical structures of the compounds used. The article presents significant, experimentally confirmed evidence of the biological effects of preservatives. I recommend publishing the article with minor corrections and additions.
Author Response
Please find enclosed the revised manuscript cells-2279041 entitled “Effect of commonly used cosmetic preservatives on healthy human skin cells” by Patrycja GÅ‚az, Agata RosiÅ„ska, Sylwia Woźniak, Anna Boguszewska-Czubara, Anna Biernasiuk and Dariusz Matosiuk. We changed the manuscript substantially as suggested by the Reviewers.
Below are our answers:
Today, the use of cosmetic products accompanies our lives from the cradle to the grave. It is therefore highly desirable to have a thorough knowledge of all their chemical, physicochemical and toxicological (cytotoxicity) properties. The manuscript is very useful in this respect. Parabens are often used in small amounts as preservatives in cosmetics, pharmaceuticals, foods, and beverages since the 1920s. Researchers have identified parabens as chemicals that can potentially disrupt hormonal systems in both males and females. Some patients were found to have a sensitivity, allergic reactions to parabens. The European Union banned isopropylparaben and isobutylparaben from use in any personal care products in 2015 until more research can be done. Diazolidinyl urea (please, add its newly determined structure) and its chemically related to imidazolidinyl urea acts as a formaldehyde releaser. Both are skin allergens.
- The article is well written, however I have a few comments. I do not recommend the use of any abbreviations in the abstract, which are not explained (ECM, or GSB and MMP-2).
Thank you for your comment. We agree with the Reviewer. Now the abbreviations are explained. Version with explanations of abbreviations should become clearer to the reader.
- The text should consistently spell out Methyl paraben vs. methyl paraben (p. 14 etc.).
Thank you for your remark. We standardized the terminology of used compounds.
- A reader with a basic science background would welcome the chemical structures of the compounds used.
Thank you for your advice. We added the table with the structures of every used compound in the study.
- The article presents significant, experimentally confirmed evidence of the biological effects of preservatives. I recommend publishing the article with minor corrections and additions.
Thank you for your kind comment. We hope that our article after the necessary corrections could be valuable for the readers.
Revisions to the manuscript were marked with the "Track Changes" function. The manuscript has undergone English revisions.
We would like to thank the Reviewers for their time and all their comments. We would greatly appreciate it if you could consider publishing the manuscript with relevant corrections and additions.
We wish to join these following statements:
We confirm that neither the manuscript nor any parts of its content are currently under consideration or published in another journal.
All authors have approved the manuscript and agree with its submission to Cells.
We look forward to hearing your comments and decision concerning the publication of our manuscript.
Best regards
Patrycja GÅ‚az & Dariusz Matosiuk
Corresponding author:
Dariusz Matosiuk
Medical University of Lublin, Chair and Department of Synthesis and Chemical Technology of Pharmaceutical Substances, 4A Chodzki Str., 20-093 Lublin, Poland.
E- mail: dariusz.matosiuk@umlub.pl
Tel: +48 81 448 72 72
Round 2
Reviewer 1 Report (New Reviewer)
-In the introduction section it is correct define for the first time the acronyms used. However, when the manuscript is very long and the acronym is not common, it can be defined again when the acronym is used later.
-The authors should check English grammar in the manuscript.
-In the figure 3, the panels shown should be delimited correctly. If possible, please add a scale on the photomicrographs.
-Please add in the figure legends, the n analyzed by group or indicate that the results were in triplicate.
-The authors should add the relevant limitations of this study that they were not considered in the research design.
Author Response
Lublin, 27.03.2023
Dear Reviewer,
please find enclosed the revised manuscript entitled “Effect of commonly used cosmetic preservatives on healthy human skin cells” by Patrycja GÅ‚az, Agata RosiÅ„ska, Sylwia Woźniak, Anna Boguszewska-Czubara, Anna Biernasiuk and Dariusz Matosiuk according to the referees’ comments (minor revisions).
Below are our answers:
In the introduction section it is correct define for the first time the acronyms used. However, when the manuscript is very long and the acronym is not common, it can be defined again when the acronym is used later.
Thank you for your valuable comment. We defined the unknown acronyms in the introduction section.
-The authors should check English grammar in the manuscript.
Thank you for your comment. English grammar in the manuscript was checked again.
-In the figure 3, the panels shown should be delimited correctly. If possible, please add a scale on the photomicrographs.
Thank you for your comment. The panels in figure 3 were changed. The scale could not be added to the photomicrographs (Table 5). We added magnification to the figure legend.
-Please add in the figure legends, the n analyzed by group or indicate that the results were in triplicate.
Thank you for your comment. We added a number of replications in the figure legends.
-The authors should add the relevant limitations of this study that they were not considered in the research design.
Thank you for your valuable comment. Below is a paragraph added to the discussion section.
The findings of this study have to be seen considering some limitations. The first is the viability test (MTS) which is only a preliminary test. Conducting proliferative marker measurement experiments could be beneficial in a broader assessment and would undoubtedly lead us to more complex conclusions. We would like to focus on the most curious compounds such as germalls. It could fill the literature gap we noticed in this topic. The second limitation concerns the amount of collagen produced. In future studies, we aim to address the test of whether collagen gene expression is regulated. Our team conducted a series of experiments on a large group of compounds. In the future, we would like to perform more detailed research on selected compounds. This would allow us to determine which of them has the most prospective application.
Minor revisions to the manuscript were marked with the "Track Changes" function. The manuscript has undergone English revisions.
We would like to thank the Reviewers for their time and all their comments.
Patrycja Glaz, Dariusz Matosiuk
This manuscript is a resubmission of an earlier submission. The following is a list of the peer review reports and author responses from that submission.
Round 1
Reviewer 1 Report
Major:
The authors do not provide any information about the dermal penetration of the investigated substances. Thus, I do not know that the used concentrations of tested substances in cell culture experiments are realistic or too low/high.
In contrast to your conclusion, it seems that grapefruit oils are not very effective against bacteria and against fungi higher concentrations than e.g. methylparabens are necessary. Why are there use as preservatives, in particular, when they could be toxic?
The toxicity relation of the Germalls are very strange and make no sense. Here, it is crucial to test with other methods than MTS, which is may be compromised by Germalls.
The experimental setup is not sufficient. When you discuss extensively methyl paraben in the discussion, why you did not used as comparison in your collagen/MMP2 experiments? So you could have a comparison to other studies and you can relate the achieved results with the tested substances.
I have lot of problems to interpret the collagen/MMP2 experiments using toxic doses (100 µg/m, sometime 25 µg/ml).
In particular, you did not relate the achieved results with the number of viable cells. Bright fields pictures and MTS are not enough and in my opinion, for example commercial grape fruit oil killed nearly all the cells. That the few cells remained (possible damaged) are able to release MMPs and collagen (figure 3) in the same extent as the control is not plausible, indicating some experimental problems. Here, you monitored maybe the release during the dying process. A better approach would be remove all media replace it and perform measurements on the next days. Thus,also here the number of viable cells and metabolism, the protein concentration of cells pellet (for western blot) under the same experimental conditions are crucial.
Conclusion:
Minor:
Method: Details primary antibodies
Table 1: too big, please separate in 3 Tables like funghi, gram posivite and gram negative.
Legend of table: Please indicate units µg/mL
Please highlight significant values with high antimicrobial efficacy (bold).
Figure 1: Do you have an idea why Germail 115 and Germail II do not have a clear dose/effect ratio?
Figure 1 and Figure 2 need a lot of space. Please try to make the figure more compact, there is a lot of space between the graphs and the titles (e.g. GSB preservative ) could be within the graph. Furthermore, the axis need more subdivision.
3.4. Influence of preservatives on collagen synthesis
Please provide information that you have only tested the effects of 4 substances on collagen synthesis and maybe repeat the reason why only this 4 and not others:
The results of Sircol assay were more and less the same, also for cells treated with toxic doses.
In my opinion here the assay failed. Do you have performed experiment without cells as negative control/blank?
As I remember, the Sircol assay need a different FCS concentration and also it was recommend to add Vitamin C /Ascorbate to cells for collagen production/release. As positive control the addition of TGF-ß should boost the collagen production. Also 24 h incubation time could be to short to accumulate collagen. Furthermore, you need to normalize on cell number, in particular, when you expect that the substances could be toxic. Thus, a nomarlization on MTT values or stained nuclei or viable cell are necessary.
Figure 3 Please mark the different panels with bold letters, you starting with a graph without indication, the second one is with (a) and the third is with (b) and then separed in (c) (d) and (e). That’s quite confusing. Please indicate directly on one side of the westenblots the shown protein, e.g. collagen for (c) and so one.
In figure 3 what is (e)? In the legend you tell something values + SD. I can only some bands. Please correct and insert the missing information/graph/values
3.5 MMP-2 activity
In 2.6 you described that protein concentrations in the sample medium was measured.
What was the result? Are there different? In particular, when toxic effects by Germal II are present it is important to know, if dead cells can cause higher protein concentration by realizing intracellular proteins, or they did not secret nothing in the experiment period because there are busy with dying. Do you used them to normalize the MMP activity.
Here, I wonder that with Germal II 100µg/ml a signal signal/activity can be observed, whereas less toxic Germail 115 caused lower MMP2 activity.
In this context, did you measured MMP2 protein by western blot? Less protein, less activity? Or did you investigate the TIMPs? Which also inhibit the MMP activity.
Reviewer 2 Report
Please refer to the submitted file.
